# Low-Abundance Protein Enrichment for Medical Applications: The Involvement of Combinatorial Peptide Library Technique

**DOI:** 10.3390/ijms241210329

**Published:** 2023-06-19

**Authors:** Egisto Boschetti, Pier Giorgio Righetti

**Affiliations:** 1JAM Conseil, 92200 Neuilly-sur-Seine, France; 2Department of Chemistry, Politecnico di Milano, 20133 Milan, Italy; righetti.pg@tiscali.it

**Keywords:** proteomics, enrichment, medical applications, low-abundance proteins

## Abstract

The discovery of low- and very low-abundance proteins in medical applications is considered a key success factor in various important domains. To reach this category of proteins, it is essential to adopt procedures consisting of the selective enrichment of species that are present at extremely low concentrations. In the past few years pathways towards this objective have been proposed. In this review, a general landscape of the enrichment technology situation is made first with the presentation and the use of combinatorial peptide libraries. Then, a description of this peculiar technology for the identification of early-stage biomarkers for well-known pathologies with concrete examples is given. In another field of medical applications, the determination of host cell protein traces potentially present in recombinant therapeutic proteins, such as antibodies, is discussed along with their potentially deleterious effects on the health of patients on the one hand, and on the stability of these biodrugs on the other hand. Various additional applications of medical interest are disclosed for biological fluids investigations where the target proteins are present at very low concentrations (e.g., protein allergens).

## 1. Introduction

In most biological fluids or extracts, proteins are present in a large number and their individual concentration difference can span over several orders of magnitude. Typical examples are given by blood serum where the dynamic concentration range is estimated at 12–15 orders of magnitude with albumin which is a largely dominant protein [1]. However, this is not the only biological sample presenting such a characteristic. The lysate of red blood cells is another example where the local concentration of the main protein, hemoglobin, is about 300 mg/mL (representing approximately 95% of the total proteome) and the least concentrated proteins are well below the ng value [2]. Plant fluids and extracts are not in a very different situation since the most abundant protein by far in leaves is RuBisCO (ribulose-1,5-bisphosphate carboxylase/oxygenase) [3] that represents more than 40% of the total protein mass and in seeds extracts are storage proteins such as β-conglycinin and glycinin accounting for about 80% of the proteins [4]. This situation creates difficulties when considering the detection and identification of the most dilutes polypeptides.

Proteins that are of high abundance mask the signal of many others of very low abundance escaping thus detection by current available instrumentation and analytical methods. This is why the exploration of low-abundance proteins (LAP) cannot be performed directly, but rather using dedicated tricks. Chemical depletion, immunosubtraction, affinity depletion, molecular filtration and precipitation are the most common. Thus, the perfection of methods allowing the detection of low-abundance proteins is essential for targeted applications. They allow for establishing a better knowledge of proteomes and the understanding of the protein interactions. The detection of protein traces also helps the design of purification procedures of isolated biopharmaceuticals such as, for instance, purified vaccines or recombinant monoclonal antibodies. Spontaneous or induced subtle modifications of protein expression are also another aspect where reaching low- or very low-abundance proteins is critical.

This review is mostly dedicated to one of the major enrichment methods based on the use of combinatorial peptide ligand libraries.

## 2. Current Methods for Low-Abundance Protein Enrichment

The large dynamic concentration range of most protein extracts is a hard technical obstacle when attempting to visualize and identify proteins of low abundance. The number of proteins present is extremely large and can reach several thousand or more. In addition, their individual concentration is very different; among a few concentrated proteins are a large number of others with a concentration close to ng/mL or lower. Useless to say that with such a low concentration it is almost impossible to detect all proteins at once for at least two reasons: (i) the capabilities of most current analytical methods are restricted to about 4–5 orders of magnitude and (ii) the signal of high abundance proteins obscures the signal of the most dilutes forms. It is within this context that the interest of scientists is evidenced by the appearance of a growing number of published reports as illustrated in Figure 1.

To circumvent this difficulty, the most obvious option is to eliminate the high-abundance protein (HAP) group. Historically, this approach started in 2003 with Pieper et al. [5] with the proposal of using a solid phase carrying few immobilized antibodies. This approach was intended to subtract a few of the most abundant proteins from human serum. Later on, immunosubtraction has been extended to a larger number of plasma proteins [6] up to about 60 [7] with a relatively modest advantage compared to the initial proposal. Close to the advantage of removing high-abundance proteins, problems have been identified such as massive co-depletion phenomena of unexpected other proteins. In some cases, the co-depletion was so massive at a point that the number of removed proteins was larger than what was found in the depleted sample [8]. In immunosubtraction, another major drawback observed is a dilution effect that, without a problematic concentration step, renders even more complicated the detection of most low-abundance proteins. Moreover, this enrichment process involving antibodies is so specific that can only be applicable to restricted biological samples. Alternative methods have been described for the enrichment of protein groups such as glycoproteins [9] and phosphoproteins [10]. Glycoproteins are the result of post-translational modifications with the addition of various glycan structures conferring singular properties of recognition and other functions. Representing a large diversity and number, they constitute a homogeneous group comprising high- and low-abundance species. They have aroused particular interest insofar as certain unexpected antigens are expressed in carcinogenesis processes and may constitute a possible path toward the discovery of biomarkers of interest [11]. Thus, dedicated glycan-based enrichment technologies are applied, such as affinity adsorption on immobilized lectins [12] and boronate-based affinity ligands [13]. 

Concerning phosphoproteins, about 30% of human proteins are phosphorylated; they participate in many biological processes such as signaling and many of them are of low abundance. Their detection, identification and analytical determinations are challenging; thus, they constitute a group of products where it makes sense to find methods of enrichment. Various approaches have been described such as hydroxyapatite adsorption [14], metal chelating chromatography [15] and immunoaffinity chromatography [16] with specific antibodies. The adsorption on these solid phases is not always bispecific and consequently contributes to the co-adoption of other proteins. Nevertheless, the associated concentration effect of these methods contributes to detecting low-abundance phosphoproteins by mass spectrometry. 

In essence, to enrich for LAP four main approaches are described: fractionation methods such as liquid chromatography, precipitation, subtraction of high-abundance proteins, targeted capture of low-abundance proteins and reduction of dynamic concentration range by CPLLs (combinatorial peptide ligand library). The main characteristics of each method are summarized in Table 1.

These enrichment method proposals have been described for animal proteomes [17], plant proteomes [18,19] and even recombinant therapeutic proteins when attempting to detect contaminating proteins from the host cells. In this domain, dedicated methods are described. They are the depletion of the largely dominant recombinant protein [20], the immunoprecipitation of protein impurity traces [21], cutoff filtration [22], size exclusion [23], various chromatographic procedures and CPLL. These approaches are described in a few published application reviews [24,25,26,27]. 

## 3. Combinatorial Peptide Ligand Library Technology

The enrichment process is in essence a separation method able to isolate a group of proteins from a large ballast of other polypeptides. The group of proteins in question must, thus, comprise a common property or similarities in structure exploited for the fractionation process. The removal of a large proportion of proteins from the sample greatly facilitates further analytical operations. The major feature of the CPLL-based procedures is not the separation of different species of proteins but rather the reduction of the concentration of the most represented proteins while increasing the concentration of proteins of low abundance [25].

Since the development of the CPLL technology, thousands of scientific publications report data on proteomics investigations when low-abundance proteins are the objective of the study. The diversity of captured protein, the diversity of biological fluids and the diversity of species (extracts from animals, plants, and bacteria) are witnesses of the large applicability of the principle as described below. This technology demonstrated a relatively easy way to identify proteins that are normally undetectable without any enrichment effect.

The solid phase combinatorial peptide library is a mixed bed of affinity sorbents for the capture of proteins. Each bead supports millions of copies of a unique hexapeptide structure made using combinatorial synthesis. Considering that for the synthesis of hexapeptides 16 natural amino acids are used, the library comprises a population of linear hexapeptides amounting to 16–17 million different structures. Thus, in principle, an appropriate volume of beads contains enough ligand partners to interact with just about each protein present in a complex proteome. The density of these ligands is about 40–60 µmol per mL of bead volume.

When CPLL is put in contact with a protein extract or a biological fluid, proteins are adsorbed by affinity interaction on their corresponding hexapeptide beads. This reaction occurs up to the bead saturation. Saturation is very rapidly reached with high-abundance proteins while for the very dilutes ones (the low-abundance proteins) it is only reached when the sample volume is large enough to meet the bead binding capacity. In that way, the amount of high-abundance proteins captured by the beads is limited as a consequence of bead saturation; the excess of these abundant proteins remains as a consequence in the supernatant. On the contrary proteins of low abundance are progressively concentrated as long as the offered volume of sample to the solid phase is increased.

The interaction between the captured protein and the hexapeptide ligands is affected by the composition of the medium. Most generally, the process of protein loading is conducted under physiological conditions. On the contrary, the protein collection is operated by using chemical agents capable to annihilate the molecular interactions. All proteins captured are thus collected for further analysis. Several possibilities have been described over the years [25] for conducting a single desorption step or a sequential elution mode. An alternative to the elution procedures is the digestion of captured proteins directly on beads and collected peptides analyzed by mass spectrometry [28,29]. The on-bead protein trypsination prior to mass spectrometry was recently further optimized and improved by extending the digestion time and also by making a pre-digestion with Lys-C endopeptidase [30] producing a large number of peptides from each protein. It is here important to say that even if some peptides are still strongly attached to the beads or lost after digestion, the mass spectrometry analysis results are not degraded because the minimum number of peptides needed for reliable protein identification is limited to two. The increased number of detected proteins, the lower risk of protein losses due to too hard elution conditions and the simplified protocol with better reproducibility, should play in favor of the on-bead protein digestion option, especially when looking for protein markers discovery and for the detection of protein impurities present in biopharmaceuticals.

The enrichment process based on solid phases combinatorial peptide ligand library has been repeatedly compared to the popular immunosubtraction where applicable, especially in proteomics investigations involving human serum or plasma. Frequently, the advantage of the number of gene products discovered has been attributed to CPLL [31,32,33]. Recently, a paper comparing several approaches has been published where experimental results indicated remarkably better performance than immunosubtraction based on 14 major antibodies [34]. Figure 2 reports the two-dimensional electrophoresis results of one comparative experiment on human serum.

Spectacular demonstrations of CPLL performance are given by three pioneering papers. The first was centred on human plasma proteins [35] where the number of gene products found after low-abundance proteins enrichment with CPLL reached 3869, a number much larger than what was known earlier in this domain at these early days of proteomics investigation and with a remarkable reproducibility. 

A second report involved the findings in red blood cell proteins [36]. In this particular case, the number of proteins found was 1578 while what was known before was limited to a few hundred. This was the first ever deep exploration of such a proteome starting from a cytoplasmic fraction of a highly purified human red blood cell preparation where unexpected minor proteins were identified. 

A third significant example is reported with the exploration of hemolymph proteins from *Limulus polyphemus*, a living fossil arthropod from the east coast of North America [37]. The direct proteome analysis by two-dimensional electrophoresis evidenced 200 protein spots while after treatment with CPLL 890 protein spots were present throughout a pH range of 3–10 and a molecular mass between 8 and 240 kDa. The following mass spectrometry analysis gave an extremely large number of proteins (around 7500), most of them unmatched because they could not be attributed to gene products due to the absence of proper annotations. A few years after, another study [38] identified 1100 unique proteins with a good confidence value upon treatment of the hemolymph with CPLL.

## 4. Identification of Early-Stage Biomarkers of Human Diseases

It is a general consensus that the discovery of early biomarkers allows for better management of diseases based on the concept that a therapeutic treatment may be more effective at the initial stage of a given pathology [39]. This is particularly true for pathologies having a low survival rate. Within this context, the major obstacle to overcome is to devise a discovery method for relevant biomarkers when their expression is still extremely limited. Their detection is very challenging because of the massive presence of all other current abundant proteins that mask the signal of very dilute disease-related proteins. It is really like “looking for a needle in a haystack”, as indicated earlier [40].

A single biomarker (up- or down-regulated) is rarely pertinent for the formal designation of a pathology; hence, it is generally admitted that the use of combinations of several of them would improve the predictivity of a given disease. The more numerous the biomarkers considered as a whole, the better the diagnosis value; however, even this approach is not a full guarantee of a correct diagnosis without adding other biological variables such as sex, age and predispositions. 

The question is how to approach the search for early biomarkers. One way would be to target a given organ where the disease is presumably located with the presumption that the misregulated expression of pathology-related proteins occurs and where their concentration is supposed to be significant [41]. This principle has been used a number of times with interesting results. Nevertheless, high-abundance proteins are still present, perturbating the detection of potential biomarkers. 

Among other interesting approaches is targeting circulating exosomes in biological fluids. They are about 100 nm membrane vesicles released by mammalian cells, including malignant ones, capable to transfer proteins in charge of cellular communication functions. They are potential reservoirs of protein markers of interest, especially when they come from malignant cells. Although of high interest, this option is quite laborious because it is contingent upon the isolation of these vesicles among numerous other similar bodies [42]. The interest of such an approach has been shown with urinary exosomes where the presence of a few low-abundance antigens was directly correlated with prostate cancer [43]. Nonetheless, other authors are less optimistic, indicating that in spite of useful clinical information, results are still preliminary and not fully conclusive [44]. 

At early disease stages of protein expression, the signs of proteome difference between a disease against a control are still extremely hard to recognize. Here, the enrichment methods are essential to visualize the weak signs of modified protein expression. Since the beginning of its development, CPLL technology was recognized as a confident and reliable approach for biomarker discovery. To evidence the presence of a potential biomarker, a direct comparison between a pathological situation must be made against a control (see scheme in Figure 3). Variations are possible as extensively described [25] according to the objectives of the application with many discovered biomarkers. It is out of the scope of this paper to make an exhaustive list of the described applications; thus, only a few representative examples of major diseases, such as cancers, are summarized first. 

Identifying early biological signs to improve the survival rates of certain cancers is critical. This is the case for pancreatic tumors, a highly invasive malignant disease with a preponderant lethal outcome [45]. The most common biomarker used is the carbohydrate antigen CA 19–9, but it has insufficient ability to detect pancreatic cancer because of its poor specificity and sensitivity. Relevant papers have reported the early discovery of pertinent markers using combinatorial peptide ligand libraries using a CPLL-treated supernatant of pancreatic cancer cultured cells [46]. Among differentially regulated proteins found, six of them were associated with survival. The most relevant were glucagon-like peptide-1, apolipoproteins CII and CIII and zinc-alpha-2-glycoprotein. 

Another aggressive cancer is HCC (hepatocellular carcinoma) [47] for which only limited therapeutic options are currently available. The biochemical diagnosis is classically based on the determination of the level of AFP (α-fetoprotein), a non-reliable marker in terms of sensitivity and specificity, present in many other cancers. A promising advancement is the detection of early-stage markers using enrichment methodologies. As suggested by Mustafa et al. [48], a biological sample (e.g., blood serum) treated with CPLL contributed significantly to this undertaking. The enriched serum using CPLL is resolved by 2D-DIGE (two-dimensional difference gel electrophoresis) and differentially expressed proteins identified by mass spectrometry, followed by a targeted quantitation using SRM (selected reaction monitoring) method. From a quite large study [49] using CPLL followed by analysis 2D (two-dimensional) electrophoresis, 4000 spots have been investigated. Among the 24 misregulated proteins found, the authors centred their attention on ApoA1, which is down-regulated by 2.45 fold. The reason for this choice was its known ability to suppress the expression of adhesion molecules and to inhibit tumor necrosis factor-α, two important factors of cancer development. No good correlation was found with AFP, as described above. This study suggests monitoring ApoA1 to follow the effectiveness of therapeutic treatments. Beyond cancers, other pathological situations are at the centre of early biomarker discovery of diagnosis/prognosis interest. Significant recent examples starting from various body fluids are illustrated hereafter.

Blood serum is the most investigated biological fluid due to its continuous circulation throughout the body from where it is supposed to collect all possible proteins from various organs, including low-abundance protein markers expressed within localized organs. Unexpected applications are progressively published. An example is given by the identification of prognostic markers of severe COVID-19 [50] from CPLL-treated serum samples from infected patients (see below for more details).

Another serum-based biomarker discovery worthy of interest is exemplified by RA (rheumatoid arthritis). Upon CPLL serum treatments at least two routes have been followed to the identification of specific signatures of the pathology. According to the classical way of identification of biomarkers, more than two dozen proteins were differentially expressed. Some were down-regulated, such as paraoxonase/arylesterase 1, proteoglycan 4 and plasminogen, while others were up-regulated, such as, for example, apolipoprotein E, thrombospondin-1 and ficolin-2. The latter represented a serious potential marker for diagnostic applications. Dedicated quantifications by ELISA assay (enzyme-linked immunosorbent assay) indicated the extent of the quantitative expression modification [51].

The second route followed for the discovery of specific signatures of RA is based on a strong presence of antibodies against citrullinated proteins that appear long before the development of the pathology. On these bases, a study has been established to determine the presence of citrullinated autoantigens among the pool of overexpressed proteins [52]. Serum samples from patients were subjected to treatment with CPLL and protein abnormalities were determined by mass spectrometry. A set of 135 misexpressed proteins was found, 11 of which contained citrulline. Exposed to anti-citrulline antibodies in patients with RA, autoantibodies against citrullinated lipopolysaccharide-binding protein were found to be particularly increased. This study concluded that the presence of anti-lipopolysaccharide binding protein antibodies could become markers not only for the diagnosis of rheumatoid arthritis, but also for defining its severity.

Still, within the discovery of markers of importance, atherosclerosis, a devastating cause of cerebrovascular and cardiovascular events, has been considered a subject of investigation. To get a global view of protein expression including low-abundance proteins several projects were centred on the use of CPLL as a means to enrich rare proteins. Enriched carotid tissue protein extracts showed important differences in low-abundance proteins between non-complicated and complicated plaques [53]. Major alterations were essentially found for aldehyde dehydrogenase, heat-shock protein 27, protein kinase C delta-binding and moesin. More recently, protein signatures of atherosclerotic plaques have been searched in the blood [54]. It appeared that the complexity of plaque rupture as the initial cause of stroke could probably be due to several proteins. A set of 76 proteins have been identified while analyzing unstable atherosclerotic plaques. Part of such a panel could be considered as prognostic signs of plaque instability.

On another subject, to predict the outcome of osteotomy and to prepare for the best treatment, dedicated protein multi-markers have been identified from CPLL-treated synovial fluid. Findings were defined as very useful to help classify patients suited to treatment with osteotomy [55].

Other body fluids such as saliva [56], urine [57] and cerebrospinal fluid [58] have been explored repeatedly with and without enrichment effects in order to detect low-abundance protein markers with benefits related to outcome and to determine the best therapeutic treatment before the advent of critical situations.

## 5. Detection of Protein Impurity Traces from Recombinant Biopharmaceuticals

Several bioproducts for therapeutic applications are presently produced by recombinant procaryotic or eucaryotic cell cultures. The most representative are monoclonal antibodies. All these recombinant products need to be purified in order to eliminate the components from either the cell culture medium or from co-expressed proteins, including proteins from the cell lysis during their life cycle. While purification processes are very efficient, some impurities are always present in trace amounts in the purified protein. The nature of these impurities is very diverse and can adversely affect the stability of recombinant proteins and induce immunogenic reactions in patients. These protein impurities called HCP are tolerated all together within the limit of 100 ppm [59], but this figure does not give a good picture of the reality because it covers an unidentified number of HCP. Protein impurities may not always be the same and their relative concentration may be different even if the total amount is similar. Commonly, quantification is performed by global immunochemical assays using polyclonal antibodies against all host cell proteins [60]. While formal identification is a priority objective, it comes up against their very low concentration and the massive presence of the main recombinant therapeutic protein. Very often the dynamic concentration range spans over several orders of magnitude, which suggests using enrichment techniques developed in proteomic studies. Various approaches have been proposed; the most current are the removal of the recombinant protein [20,61,62], the immunoprecipitation of HCP [63], molecular size discriminating methods (cutoff filtration [22], size exclusion [23]) and various chromatographic procedures. In addition to these general enrichment technologies, CPLLs appear to be a very effective way to enhance the presence of low-abundance proteins while largely diminishing the recombinant dominant recombinant protein (see previous sections). A pioneering work on this principle was published in 2006–2007 and demonstrated the usefulness of highlighting the presence of host impurities in highly purified recombinant proteins [64,65] (see examples in Figure 4). It took several years before it was formally applied as a solution to concrete problems such as biodrug stability and patient safety. In a demonstration example [66], a recombinant protein expressed in CHO (Chinese hamster ovary) cells was spiked with several foreign proteins in a range of concentration between 10 and 1000 ppm. The sample was submitted to LC-MS/MS analysis before and after treatment with CPLL. While no spiked proteins were detected in the control, the CPLL-treated sample showed all the spiked proteins and, in addition, revealed the presence of 30 other host cell proteins with an enrichment factor estimated between 80 and 700 fold.

In another study [63], the enrichment with a similar technology reached 1000 fold from a sample of purified monoclonal antibodies. When applied to a commercial recombinant monoclonal antibody, 527 proteins from engineered CHO cells appeared with an enrichment rate of 100–400 fold.

In an attempt to improve the effectiveness of the enrichment technology, the capture process of HCP from monoclonal antibodies has been optimized by limited trypsin digestion [67]. Here, the number of identified proteins was in the best case 850 with a reproducibility of more than 80%; the enrichment was up to 7694 fold with the capability to detect 0.05 ppm.

More recently [68], the enrichment of foreign proteins from the host cells by CPLL was followed by an improved method of quantitation. It is actually important to have accurate quantitative data about certain HCP that are particularly detrimental to biodrug stability and have the certainty that they are present at a concentration below a critical level. This is the case for several degrading enzymes, such as esterases, thioesterases, lipases and carboxypeptidases, that could be reliably quantified to sub-ppm level with good accuracy and precision.

At this stage, the detection, identification and correct quantitation of each critical protein impurity can become a current practice with the support of enrichment techniques.

## 6. Discovery of Low-Concentration Allergens

Allergy is an immune response of an organism against a foreign molecule called allergen. The latter are usually proteins of different origins that can activate a cascade of events producing undesirable effects that can go as far as the death of the organism. Not all heterologous proteins are recognized by the body as allergens, but when they are, there is a specific interaction with the body’s immunoglobulin E with consequent reactions. Without going into the details of the biochemical mechanisms, it is important to mention here that there are no well-identified allergenic peptide structures, although the replacement of an epitope responsible for an allergic reaction can be neutralized just by the replacement of an amino acid [69]. While IgE can recognize certain epitopes of foreign proteins without inducing allergic reactions [70], there are situations where the same IgE recognizes proteins of different origins that share the same sequential epitope of an allergenic antigen. This is the reason why there are common families of allergens from different species [71].

There are allergens of plant origin (e.g., pollen, seeds, fruits), animal allergens (mites, fish) and certain components of biological fluids such as milk and eggs. The number of protein allergens is constantly increasing in relation to the improvement of technologies capable of detecting those of low abundance. It is in this context that enrichment methods become of interest [72,73].

While the quantitation of allergens is operated by immunochemical methods (e.g., ELISA) [74], the discovery of new protein allergens is mostly performed by immunoblot techniques [75]. CPLL and ELISA methodologies are not competitive but, rather, complementary techniques since the former allows novel biomarker discoveries and the latter is then designed for their quantification. In short (Figure 5), the protein extract that may contain an unknown allergen is fractionated by electrophoretic techniques and subjected to the serum of an identified allergic patient. IgE from the patient interacts with the allergens and forms a complex. A second reaction follows using labelled IgG against the human IgE and the super-complex obtained revealed by histochemical reactions. Once the right positioning of the allergen is found on the electrophoretic plate, it is extracted and submitted to LC-MS/MS identification. This field of investigation lends itself well to the use of enrichment methods, in particular using CPLL.

Allergens from animal sources are probably less representative than those from plants. The most common animal allergens are from milk, eggs, fish and sea products, and small organisms such as arthropods and mollusks. The CPLL treatment of animal biological extracts contributed to the identification of several allergenic proteins from native [76] and sterilization-treated milk [77]. Eggs are also well known to comprise allergens that can be evidenced by CPLL [78]. Sometimes they are dominant proteins directly detected; in other instances, low-abundance proteins are responsible for allergic reactions and need amplification processes [76]. Among allergens discovered with the assistance of CPLL, the analysis of egg white revealed the allergenic nature of clusterin and an ovoinhibitor [78]. Beyond lactalbumin, caseins, lactoglobulin and lactoferrin, traces of polymorphic immunoglobulins have been evidenced using a serum from a selected patient [76,79]. Insect venoms are also intensively investigated for their content of allergic polypeptides that can also be of low abundance [80]. 

Plant allergens occupy a large place among current studies. They come from several distinct organs and can cause significant damage to human health. However, proteins in plants are very dilute compared to animals and allergens represent only a little fraction of them rendering their detection challenging. For easy investigations, enrichment methods applied to plant extracts are mandatory [81,82]. Among the most adopted enrichment techniques, CPLL plays a central role because of its general applicability. It has been described for the detection of allergens in various plant organs. For instance, allergens have been found in many fruits, such as bananas [83] and mango [84], where some of them are in common. Allergens from cypress pollens have recently been investigated with the adoption of CPLL enrichment techniques [85,86,87]. The analysis of enriched pollen extracts evidenced allergic low-abundance proteins such as chaperon protein HSP104, sigb-regulated protein, glyoxalase 1 and malate dehydrogenase. Interestingly, allergens from animals and plants can also be found in food by the composition ingredients. In this particular context, the presence of traces of allergen is really critical since they are frequently present as traces well below the sensitivity of current analytical methods necessitating the use of enrichment techniques [88]. A variety of food ingredients from plants and animals carrying proteinaceous allergens are currently used (seeds, nuts, fruits, wheat, flour, milk and eggs). Some of them require attention for very serious consequences on the health of consumers. This is the case with peanuts, which commonly carry allergens. Unfortunately, there are polypeptides not yet listed as allergens and their detection and identification in food is complicated by huge amounts of food matrix, masking their presence and delivering negative or false results if enrichment procedures are not used [89]. 

At present, great efforts are still ongoing to enlarge the knowledge in allergomics via proteomics studies with the help of enrichment techniques and high-throughput mass spectrometry. Although novel allergens will implement the long existing list from plants and animal biological extracts, the detection of new or known allergenic polypeptides from food remains a challenge. The detection of allergens resulting from post-translational modifications (glycation, citrullination, carbonylation and many others) is another field of investigation. On this matter, enrichment procedures accompanied by highly specific group capture methods could represent a path in which to follow. 

## 7. Other Medical Involvements

Environmental contingencies can induce physiological negative consequences whose knowledge is still limited. In this area, the analysis of gene expression can be an important step forward in prevention. This is precisely the case of space exploration by human beings or living beings in general for which it is in practice impossible to predict the influence of microgravity, exposure to various radiations and confinement. These influences operate at low and progressive levels of gene expression with adaptation or deregulation. The analysis of low-abundance proteins is one way that can provide information on the level of changes. This field of studies started before 2010 with simulations and was gradually developed. In 2014 a study was engaged on the influence of sodium chloride balance on human inflammatory processes during the Mars105 isolation period program [90]. To this end, two thousand low-abundance proteins from the urine samples of six volunteers were investigated. It has been concluded that a reduction of sodium chloride consumption probably limits the activation of an inflammatory process.

In 2019 another study was performed on a small cohort of astronauts to understand the effects of a prolonged stay in space [91]. Significant changes in proteins involved in the hemostasis system and post-translational changes, particularly in phosphorylation, have been demonstrated. The same scientific team studied the effects of microgravity on an endothelial cell culture [92] and found changes in certain elements of the cytoskeletal structure by a complex regulatory system involving Rho proteins. All these results were obtained after submitting cell extracts with CPLL to enrich low-abundance proteins. 

This type of study was extended to human cells grown in microgravity [93] and protein extracts treated with CPLL before analysis. The results confirmed the previous conclusions with an increase in filamin-A, alpha-actinin and myosin light polypeptide 6. It was also found that other mechanisms were involved at the cytoskeleton level and adhesion phenomena. Long stays in space also induce general oxidative stress, altering several biological functions with modified protein expression with an increased risk of thrombosis, as reported [94] and evidenced by enrichment processes. 

In another medical domain and closer to the recent reality, articles reported the use of CPLL along with immunosubtraction to evidence protein expression differences with the identification of prognostic biomarkers of severe COVID-19 [50]. More than two dozen differentially expressed proteins were found implicated in cardiovascular disorders and inflammation mechanisms. Acid-labile subunit of insulin-like growth factor binding protein and chitinase-3-like protein 1 were found as powerful prognostic markers contributing to providing adapted therapeutic treatments. In this hot domain, various experimental data have been produced thanks to the CPLL enrichment effect, such as the identification of biomarker candidates of acute respiratory distress syndrome contingent upon COVID-19 infection, which is useful for specific therapeutic targets [95]. Prior to analysis here, the blood serum was either submitted to an immunosubtraction with 14 multiple affinity removal systems or to enrichment with combinatorial peptide ligand libraries. From this study, more than a dozen of mis-expressed proteins seemed involved in cytokine signaling such as tumor necrosis factor, interleukin-1β and IL-6 and implicated in systemic inflammatory processes. 

## 8. Conclusions

A multi-year survey of scientific activity around the use of low-abundance protein enrichment indicates a clear evolution of the use of CPLL technology. While adopted initially for the elucidation of proteomes, the most recent years show a clear trend toward the detection and identification of rare proteins in various other domains. In the last 8–10 years the number of publications devoted to low-abundance proteins has been constantly increasing [96] and, essentially, applied in three domains of applications. The first was and still is the detection of biomarkers of diagnostic interest that appear at the beginning of a disease not only for major pathologies such as cancer but also relative to numerous other diseases. The second application domain is to find foreign proteins from recombinant biopharmaceuticals. The third is the discovery of unknown allergens, another medical domain where the adverse effects of foreign environmental materials are constantly growing. 

These trends are not dissociated from the improvement of specific applications as repeatedly reported in the last few years [97,98,99]. Beyond the current CPLL products which have been commercially available for several years, technological developments are probably not going to be slowed down. In fact, it is anticipated that in order to make the detection and identification operations more effective and sensitive, two or more protein enrichment and complementary operations would be assembled. For example, phosphoprotein enrichment with dedicated solid phases could be followed by treatments with CPLL. In this case, even the most dilute phosphoproteins would be enriched, and thus, in a position to be easily detected. These operations are not a vision of the mind since already the first combinations of enrichment methods have been described [100,101]. Blends of simple or more complicated libraries are envisioned with better specificities for target proteins on the attention to post-translational modifications will be developed in order to better understand the influence of the environment on cell communication and adaptation. The question does not seem prevented by the difficulty of discovering unknown signatures of physiological changes, but rather to elucidate the biological significance of the protein markers that are currently discovered.

Within the domain of HCP, the standardization of recombinant cell cultures will simplify the task of finding traces of proteins. It is then foreseen that only foreign dangerous proteins, for the stability of biopharma products and the protection of a patient’s health, will specifically be defined and quantified on each production lot by dedicated assays.

## Figures and Tables

**Figure 1 ijms-24-10329-f001:**
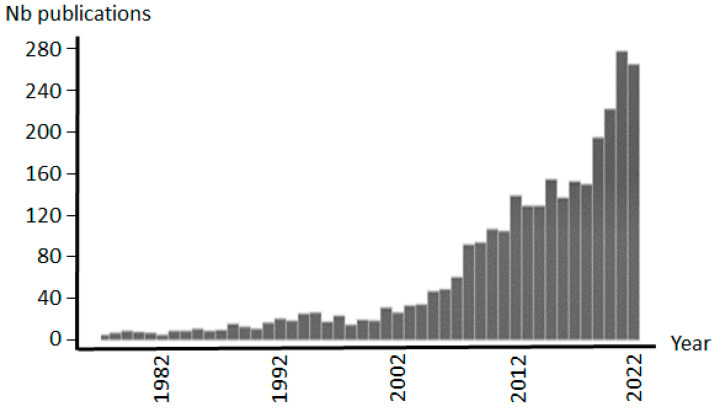
Evolution of the number of enrichment-based peer-reviewed publications over the last 40 years with more than 2600 in total. The number of scientific papers increases rapidly during the last decade. Compilation of data from PubMed (NIH, National Library of Medicine).

**Figure 2 ijms-24-10329-f002:**
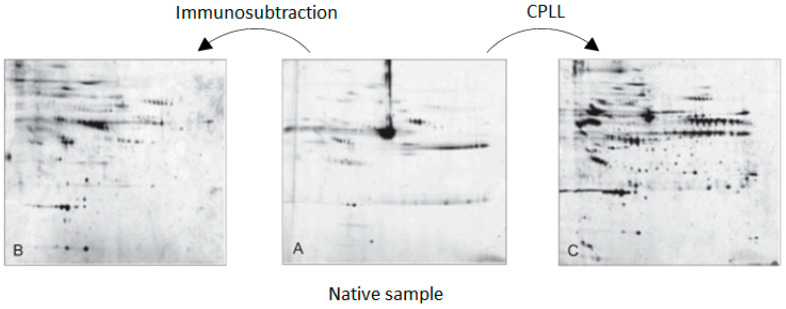
Two-dimensional electrophoresis analysis of human blood serum before (native (**A**)) and after enrichment (immunosubtraction on the left (**B**) and combinatorial peptide library on the right (**C**)). pH gradient 3 to 10; molecular mass range 250 kDa and 10 kDa; Coomassie staining). Adapted from Boschetti and Righetti [25].

**Figure 3 ijms-24-10329-f003:**
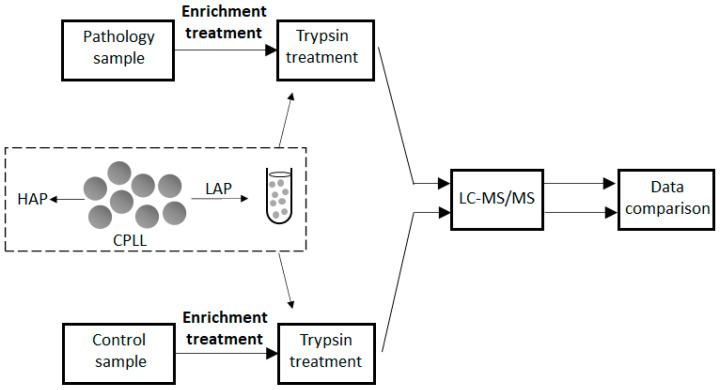
Example of bloc diagram used for the discovery of biomarkers of low abundance (early stage) with the involvement of the enrichment procedure. The sample is first enriched followed by a complete trypsination to produce peptides. The latter are then fractionated and protein identified by mass spectrometry. Comparisons are performed against a control sample. The central part of the figure (dotted frame) schematically represents the CPLL treatment of each sample. HAP: high-abundance proteins; LAP: low-abundance proteins.

**Figure 4 ijms-24-10329-f004:**
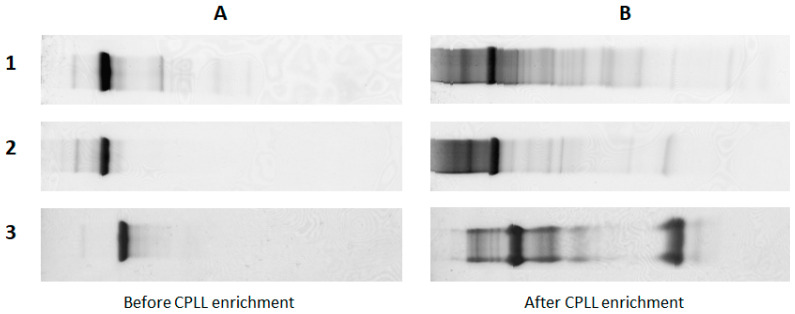
SDS-polyacrylamide gel electrophoresis of purified recombinant proteins before and after enrichment to evidence host cell protein impurity traces (for details see Refs. [64,65] from where it is adapted). (**A**) Purified recombinant proteins (1: purified monoclonal antibodies by mixed-mode chromatography; 2: purified monoclonal antibody by anion exchange chromatography; 3: purified *Staphylococcus aureus* protein A by IgG affinity chromatography). (**B**) Same protein samples, in the same order after treatment with combinatorial peptide ligand library to enrich for low-abundance proteins.

**Figure 5 ijms-24-10329-f005:**
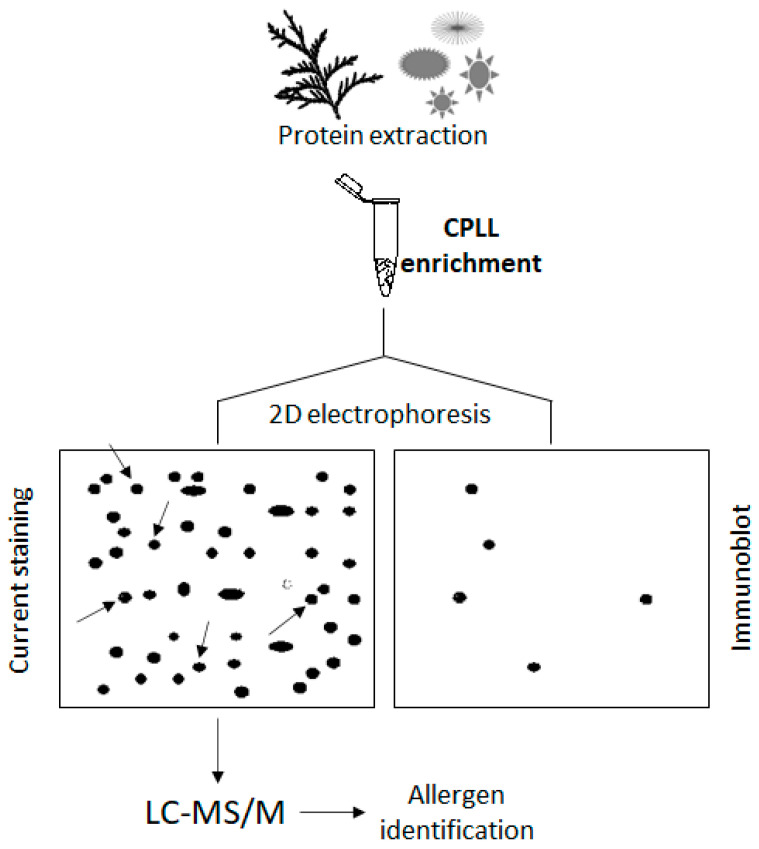
Experimental setup for the identification of allergens from a plant extract (for example, cypress pollen). After protein extraction and enrichment (e.g., by CPLL) the sample is submitted to 2D electrophoresis (two plates). One gel plate is classically stained to visualize each protein spot (plate on the left). The other gel plate (plate on the right) is submitted to an immunoblot with blood serum from an allergic patient (comprising IgE antibodies). Allergens are then visualized by immunochemical reactions. The two plates are then compared. Considered positive spots from the first plate are excised, extracted and submitted to the current procedure of protein identification by LC-MS/MS. The arrows indicate the allergen spots that are extracted for identification.

**Table 1 ijms-24-10329-t001:** Comparison of main enrichment methods used in proteomics investigations.

Method	Principle	Advantages	Drawbacks
Fractionation	Chromatography	High binding capacity Cheap Various conditions	Fraction overlapping Non specific Large dilution
Precipitation	Differential solubility	Easy handling Cheap Large and small samples Large applications	Non specific Protein entrapping Rough method Fraction overlapping
Immunosubtraction	Antibodies against HAP	High specificity Easy handling Small samples	Restricted samples Large co-subtraction Large dilution Expensive Low binding capacity
Capture of LAP groups	Various affinity ligands	Group specific Large choice Concentration of LAP	Non-specific binding Restricted to protein groups
Reduction of dynamic range with CPLL	Multiple affinity-like overloading.	Concentration of LAP Reduction of HAP No sample restriction Possible fractionated harvesting	Large samples Expensive Single use

## Data Availability

Not applicable.

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
