# Peer review of "Low-Abundance Protein Enrichment for Medical Applications: The Involvement of Combinatorial Peptide Library Technique"

_ijms, 2023, doi:10.3390/ijms241210329_

Round 1

Reviewer 1 Report

Comments on “Low-abundance protein enrichment for medical applications: the involvement of combinatorial peptide library technique.” by Boschetti and Righetti

In this manuscript, authors give an overview of combinatorial peptide ligand library technology for detection of low-abundance proteins in different fields such as biomarkers for human diseases, host cell proteins in recombinant biopharmaceuticals, plant based allergens and other medical involvements. I find the manuscript too verbose and vague at places, and highly focused on demonstrating that CPLL increased the number of proteins detected or identified. In a review of this kind, one would expect to find detailed descriptions on CPLL methodology, comparison with relevant techniques already in use and how CPLL scores better over the existing modalities, and finally future prospects for commercialization, rather than lengthy details on the low-abundance proteins themselves. Figures should focus on CPLL rather than downstream general proteomic workflow like isoelectric focusing, 2D gel electrophoresis or LC-MS/MS.

I have following queries/ suggestions to improve the manuscript.

1. Lines 115-116: Authors have already written a review on the same topic, combinatorial peptide ligand library technology, which is submitted to another journal. A copy of the review should be submitted along with this manuscript to verify that it does not overlap with this review manuscript.

2. Lines 135-142: Here, authors give specific details of the technique. Please cite appropriate references to support the data.

3. Authors should shed more light on hexapeptide based affinity enrichment. It is difficult to imagine a library comprising of a population of linear hexapeptides amounting 16-17 million different structures could enrich desired low-abundance proteins. No references are cited.

4. A schematic on CPLL and its comparison with other methods discussed in this manuscript would be highly useful for the readers.

5. Figure 2: Do the data come from author’s own work? If yes, a description on materials and methods should be provided. If no, cite appropriate references.

6. Lines 186-204: Authors have cited very old papers (refs. 33-35, 2007 – 2010 range) to provide demonstration of CPLL performance. One wonders whether the technique is still relevant.

7. Figure 3 shows a standard proteomic workflow. Authors should depict how CPLL is of help for enrichment of desired low-abundance proteins.

8. Early stage biomarkers: Authors should focus on a few success stories rather than providing exhaustive descriptions on pathology and treatment modalities. Compilation of data on CPLL guided detection of biomarkers in a tabular form would be useful.

9. Early stage biomarkers: Here, authors should compare CPLL with ELISA, as later is a routine technique for commercial applications.

10. HCPs, allergens and other medical applications: Please demonstrate the contribution of CPLL with very specific examples.

11.  Authors should shed light on prospects of CPLL for commercialization.

The manuscript is too verbose and better framing of sentences is required. Some of the strange use of words are: Outmost (Line 43), make a focus on this question (Line 50), such and such category of proteins (Lines 51-52), etc. to list a few.

Author Response

Thank you for your comments and suggestions. While answering to your questions we have without doubts significantly improved our review project.

The document attached comprises answers to all your points.

Best Regards

Reviewer 2 Report

I really appreciated reading this review, really very interesting for all the researchers involved in biomarker discovery and proteomics. It is well organized and provides a lot of useful information. The only improvement I suggest is to expand paragraph 2 (current methods for low-abundance protein enrichment). It is too concise and would require further information and the addition of other bibliographic references.

Author Response

Many thanks four your kind appreciations on our manuscript project. 

The paragraph 2 has been modified and enriched with novel references. 

The manuscript is much improved and hope it is now in conditions of publications.

Round 2

Reviewer 1 Report

Comments on revised manuscript “Low-abundance protein enrichment for medical applications: the involvement of combinatorial peptide library technique.” by Boschetti and Righetti.

While overall the manuscript look better than the original version, following issues are still pending.

1. In the manuscript file, I do not see amended figures (Figs 2-5). Authors need to address this.

2. Some of the suggested language changes not attended to. eg. "This review is intended to make a focus on"

Some of the suggested language changes not attended to. eg. "This review is intended to make a focus on"

Author Response

Thank you for the novel suggestions for improvements. 

The manuscript has been revised as requested on both corrections of language and the insertion of amended Figures within the text.

We hope the manuscript is now in position to be accepted.

With best Regards
